# Learning Non-Gaussian Multi-Index Model via Second-Order Stein's Method

**Zhuoran Yang**[*]  **Krishna Balasubramanian**[*]  **Zhaoran Wang**[†]  **Han Liu**[†]

## Abstract

We consider estimating the parametric components of semiparametric multi-index models in high dimensions. To bypass the requirements of Gaussianity or elliptical symmetry of covariates in existing methods, we propose to leverage a second-order Stein's method with score function-based corrections. We prove that our estimator achieves a near-optimal statistical rate of convergence even when the score function or the response variable is heavy-tailed. To establish the key concentration results, we develop a data-driven truncation argument that may be of independent interest. We supplement our theoretical findings with simulations.

## 1 Introduction

We consider the semiparametric index model that relates the response $Y \in \mathbb{R}$ and the covariate $X \in \mathbb{R}^d$ as $Y = f(\langle \beta_1^*, X \rangle, \ldots, \langle \beta_k^*, X \rangle) + \epsilon$, where each coefficient $\beta_\ell^* \in \mathbb{R}^d$ ($\ell \in [k]$) is $s^*$-sparse and the noise term $\epsilon$ is zero-mean. Such a model is known as sparse multiple index model (MIM). Given $n$ i.i.d. observations $\{X_i, Y_i\}_{i=1}^n$ of the above model with possibly $d \gg n$, we aim to estimate the parametric component $\{\beta_\ell^*\}_{\ell \in [k]}$ when the nonparametric component $f$ is unknown. More importantly, we do not impose the assumption that $X$ is Gaussian, which is commonly made in the literature. Special cases of our model include phase retrieval, for which $k = 1$, and dimensionality reduction, for which $k \geq 1$. Motivated by these applications, we make a distinction between the cases of $k = 1$, which is also known as single index model (SIM), and $k > 1$ in the rest of the paper.

Estimating the parametric component $\{\beta_\ell^*\}_{\ell \in [k]}$ without knowing the exact form of the link function $f$ naturally arises in various applications. For example, in one-bit compressed sensing [3, 39] and sparse generalized linear models [36], we are interested in recovering the underlying signal vector based on nonlinear measurements. In sufficient dimensionality reduction, where $k$ is typically a fixed number greater than one but much less than $d$, we aim to estimate the projection onto the subspace spanned by $\{\beta_\ell^*\}_{\ell \in [k]}$ without knowing $f$. Furthermore, in deep neural networks, which are cascades of MIMs, the nonparametric component corresponds to the activation function, which is prespecified, and the goal is to estimate the linear parametric component, which is used for prediction at the test stage. Hence, it is crucial to develop estimators for the parametric component with both statistical accuracy and computational efficiency for a broad class of possibly unknown link functions.

**Challenging aspects of index models:** Several subtle issues arise from the optimal estimation of SIM and MIM. In specific, most existing results depend crucially on restrictive assumptions on $X$ and $f$, and fail to hold when those assumptions are relaxed. Such issues arise even in the low-dimensional setting with $n \gg d$. Let us consider, for example, the case of $k = 1$ with a known link function $f(z) = z^2$. This corresponds to phase retrieval, which is a challenging inverse problem that has regained interest in the last few years along with the success of compressed sensing. A straightforward way to estimate $\beta^*$ is via nonlinear least squares regression [17], which is a nonconvex optimization problem. [6] propose an estimator based on convex relaxation. Although their estimator is optimal

---

[*]Princeton University, email: {`zy6, kb18`}`@princeton.edu`

[†]Tencent AI Lab & Northwestern University, email: {`zhaoranwang, hanliu.cmu`}`@gmail.com`

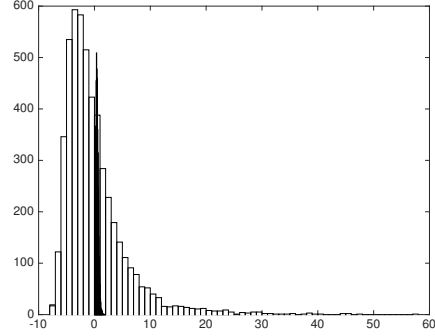

Figure 1: Histogram of the score function based on 10000 independent realizations of the Gamma distribution with shape parameter 5 and scale parameter 0.2. The dark solid histogram concentrated around zero corresponds to the Gamma distribution, and the transparent histogram corresponds to the distribution of the score function of the same Gamma distribution.

when $X$ is sub-Gaussian, it is not agnostic to the link function, i.e., the same result does not hold if the link function is not quadratic. Direct optimization of the nonconvex phase retrieval problem is considered by [5] and [30], which propose statistically optimal estimators based on iterative algorithms. However, they rely on the assumption that $X$ is Gaussian. A careful look at their proofs shows that extending them to a broader class of distributions is significantly more challenging — for example, they require sharp concentration inequalities for polynomials of degree four of $X$, which leads to a suboptimal statistical rate when $X$ is sub-Gaussian. Furthermore, their results are also not agnostic to the link function. Similar observations could be made for both convex [21] and nonconvex estimators [4] for sparse phase retrieval in high dimensions.

In addition, a surprising result for SIM is established in [28]. They show that when $X$ is Gaussian, even when the link function is unknown, one could estimate $\beta^*$ at the optimal statistical rate with Lasso. Unfortunately, their assumptions on the link function are rather restrictive, which rule out several interesting models including phase retrieval. Furthermore, none of the above approaches are applicable to MIM. A line of work pioneered by Ker-Chau Li [18–20] focuses on the estimation of MIM in low dimensions. We will provide a discussion about this line of work in the related work section, but it again requires restrictive assumptions on either the link function or the distribution of $X$. For example, in most cases $X$ is assumed to be elliptically symmetric, which limits the applicability.

To summarize, there are several subtleties that arise from the interplay between the assumptions on $X$ and $f$ in SIM and MIM. An interesting question is whether it is possible to estimate the parametric component in SIM and MIM with milder assumptions on both $X$ and $f$ in the high-dimensional setting. In this work, we provide a partial answer to this question. We construct estimators that work for a broad class of link functions, including the quadratic link function in phase retrieval, and for a large family of distributions of $X$, which are assumed to be known a priori. We particularly focus on the case where $X$ follows a non-Gaussian distribution, which is not necessarily elliptically symmetric or sub-Gaussian, therefore making our method applicable to various situations that are not feasible previously. Our estimators are based on a second-order variant of Stein's identity for non-Gaussian random variables, which utilizes the score function of the distribution of $X$. As we show in Figure 1, even when the distribution of $X$ is light-tailed, the distribution of the score function of $X$ could be arbitrarily heavy-tailed. In order to develop consistent estimators within this context, we threshold the score function in a data-driven fashion. This enables us to obtain tight concentration bounds that lead to near-optimal statistical rates of convergence. Moreover, our results also shed light on two related problems. First, we provide an alternative interpretation of the initialization in [5] for phase retrieval. Second, our estimators are constructed based on a sparsity constrained semidefinite programming (SDP) formulation, which is related to a similar formulation of the sparse principal component analysis (PCA) problem (see Section 4 for a detailed discussion). A consequence of our results for SIM and MIM is a near-optimal statistical rate of convergence for sparse PCA with heavy-tailed data in the moderate sample size regime. In summary, our contributions are as follows:

- We construct estimators for the parametric component of high-dimensional SIM and MIM for a class of unknown link function under the assumption that the covariate distribution is non-Gaussian but known a priori.

- We establish near-optimal statistical rates for our estimators. Our results complement existing ones in the literature and hold in several cases that are previously not feasible.

- We provide numerical simulations that confirm our theoretical results.

**Related work**: There is a significant body of work on SIMs in the low-dimensional setting. We do not attempt to cover all of them as we concentrate on the high dimensional setting. The success of Lasso and related regression estimators in high-dimensions enables the exploration of high-dimensional SIMs, although this is still very much work in progress. As mentioned previously, [25, 26, 28] show that Lasso and phase retrieval estimators could also work for SIM in high dimensions assuming the covariate is Gaussian and the link function satisfies certain properties. Very recently, [10] relax the Gaussian assumption and show that a modified Lasso-type estimator works for elliptically symmetric distributions. For the case of monotone link function, [38] analyze a nonconvex least squares estimator under the assumption that the covariate is sub-Gaussian. However, the success of their estimator hinges on the knowledge of the link function. Furthermore, [15, 23, 31, 32, 40] analyze the sliced inverse regression estimator in the high-dimensional setting, focusing primarily on support recovery and consistency properties. The Gaussian assumption on the covariate restricts them from being applicable to various real-world applications involving heavy-tailed or non-symmetric covariate, for example, problems in economics [9, 12]. Furthermore, several results are established on a case-by-case basis for specific link functions. In specific, [1, 3, 8, 39] consider one-bit compressed sensing and matrix completion respectively, where the link function is assumed to be the sign function. Also, [4] propose nonconvex estimators for phase retrieval in high dimensions, where the link function is quadratic. This line of work, except [1], makes Gaussian assumptions on the covariate and is specialized for particular link functions. The non-asymptotic result obtained in [1] is under sub-Gaussian assumptions, but the estimator therein lacks asymptotic consistency.

For MIMs, relatively less work studies the high-dimensional setting. In the low-dimensional setting, a line of work on the estimation of MIM is proposed by Ker-Chau Li, including inverse regression [18], principal Hessian directions [19], and regression under link violation [20]. The proposed estimators are applicable for a class of unknown link functions under the assumption that the covariate follows Gaussian or symmetric elliptical distributions. Such an assumption is restrictive as often times the covariate is heavy-tailed or skewed [9, 12]. Furthermore, they concentrate only on the low-dimensional setting and establish asymptotic results. The estimation of high-dimensional MIM under the subspace sparsity assumption is previously considered in [7, 32] but also under rather restrictive distribution assumptions on the covariate.

**Notation**: We employ $[n]$ to denote the set $\{1, \ldots, n\}$. For a vector $v \in \mathbb{R}^d$, we denote by $\|v\|_p$ the $\ell_p$-norm of $v$ for any $p \geq 1$. In addition, we define the support of $v \in \mathbb{R}^d$ as $\mathrm{supp}(v) = \{j \in [d], v_j \neq 0\}$. We denote by $\lambda_{\min}(A)$, the minimum eigenvalue of matrix $A$. Moreover, we denote the elementwise $\ell_1$-norm, elementwise $\ell_\infty$-norm, operator norm, and Frobenius norm of a matrix $A \in \mathbb{R}^{d_1 \times d_2}$ to be $\|\cdot\|_1, \|\cdot\|_\infty, \|\cdot\|_{\mathrm{op}}$, and $\|\cdot\|_{\mathrm{F}}$, correspondingly. We denote by $\mathrm{vec}(A)$ the vectorization of matrix $A$, which is a vector in $\mathbb{R}^{d_1 d_2}$. For two matrices $A, B \in \mathbb{R}^{d_1 \times d_2}$, we denote the trace inner product to be $\langle A, B \rangle = \mathrm{Trace}(A^\top B)$. Also note that it could be viewed as the vector inner product between $\mathrm{vec}(A)$ and $\mathrm{vec}(B)$. For a univariate function $g \colon \mathbb{R} \to \mathbb{R}$, we denote by $g \circ (v)$ and $g \circ (A)$ the output of applying $g$ to each element of vector $v$ and matrix $A$, respectively. Finally, for a random variable $X \in \mathbb{R}$ with density $p$, we use $p^{\otimes d} \colon \mathbb{R}^d \to \mathbb{R}$ to denote the joint density of $X_1, \cdots, X_d$, which are $d$ identical copies of $X$.

## 2 Models and Assumptions

As mentioned previously, we consider the cases of $k = 1$ (SIM) and $k > 1$ (MIM) separately. We first discuss the motivation for our estimators, which highlights the assumptions on the link function as well. Recall that our estimators are based on the second-order Stein's identity. To begin with, we present the first-order Stein's identity, which motivates Lasso-type estimators for SIMs [25, 28].

**Proposition 2.1** (First-Order Stein's Identity [29])**.** Let $X \in \mathbb{R}^d$ be a real-valued random vector with density $p$. We assume that $p \colon \mathbb{R}^d \to \mathbb{R}$ is differentiable. In addition, let $g : \mathbb{R}^d \to \mathbb{R}$ be a continous function such that $\mathbb{E}[\nabla g(X)]$ exists. Then it holds that

$$\mathbb{E}\big[g(X) \cdot S(X)\big] = \mathbb{E}\big[\nabla g(X)\big],$$

where $S(x) = -\nabla p(x)/p(x)$ is the score function of $p$.

One could apply the above Stein's identity to SIMs to obtain an estimator of $\beta^*$. To see this, note that when $X \sim N(0, I_d)$ we have $S(x) = x$ for $x \in \mathbb{R}^d$. In this case, since $\mathbb{E}(\epsilon \cdot X) = 0$, we have

$$\mathbb{E}(Y \cdot X) = \mathbb{E}\big[f(\langle X, \beta^* \rangle) \cdot X\big] = \mathbb{E}\big[f'(\langle X, \beta^* \rangle)\big] \cdot \beta^*.$$

Thus, one could estimate $\beta^*$ by estimating $\mathbb{E}(Y \cdot X)$. This observation leads to the estimator proposed in [25, 28]. However, in order for the estimator to work, it is necessary to assume $\mathbb{E}[f'(\langle X, \beta^* \rangle)] \neq 0$. Such a restriction prevents it from being applicable to some widely used cases of SIM, for example, phase retrieval in which $f$ is the quadratic function. Such a limitation of the first-order Stein's identity motivates us to examine the second-order Stein's identity, which is summarized as follows.

**Proposition 2.2** (Second-Order Stein's Identity [13]). We assume the density of $X$ is twice differentiable. We define the second-order score function $T \colon \mathbb{R}^d \to \mathbb{R}^{d \times d}$ as $T(x) = \nabla^2 p(x)/p(x)$. For any twice differentiable function $g \colon \mathbb{R}^d \to \mathbb{R}$ such that $\mathbb{E}[\nabla^2 g(X)]$ exists, we have

$$\mathbb{E}\big[g(X) \cdot T(X)\big] = \mathbb{E}\big[\nabla^2 g(X)\big]. \tag{2.1}$$

Back to the phase retrieval example, when $X \sim N(0, I_d)$, the second-order score function is $T(x) = xx^\top - I_d$, for $x \in \mathbb{R}^d$. Setting $g(x) = \langle x, \beta^* \rangle^2$ in (2.1), we have

$$\mathbb{E}\big[g(X) \cdot T(X)\big] = \mathbb{E}\big[g(X) \cdot (XX^\top - I_d)\big] = \mathbb{E}\big[\langle X, \beta^* \rangle^2 \cdot (XX^\top - I_d)\big] = 2\beta^*\beta^{*\top}. \tag{2.2}$$

Hence for phase retrieval, one could extract $\pm\beta^*$ based on the second-order Stein's identity even in the situation where the first-order Stein's identity fails. In fact, (2.2) is *implicitly* used in [5] to provide a spectral initialization for the Wirtinger flow algorithm in the case of Gaussian phase retrieval. Here, we establish an alternative justification based on Stein's identity for why such an initialization works. Motivated by this key observation, we propose to employ the second-order Stein's identity to estimate the parametric component of SIM and MIM with a broad class of unknown link functions as well as non-Gaussian covariates. The precise statistical models we consider are defined as follows.

**Definition 2.3** (SIM with Second-Order Link). The response $Y \in \mathbb{R}$ and the covariate $X \in \mathbb{R}^d$ are linked via

$$Y = f(\langle X, \beta^* \rangle) + \epsilon, \tag{2.3}$$

where $f \colon \mathbb{R} \to \mathbb{R}$ is an unknown function, $\beta^* \in \mathbb{R}^d$ is the parameter of interest, and $\epsilon \in \mathbb{R}$ is the exogenous noise with $\mathbb{E}(\epsilon) = 0$. We assume the entries of $X$ are i.i.d. random variables with density $p_0$ and that $\beta^*$ is $s^*$-sparse, i.e., $\beta^*$ contains only $s^*$ nonzero entries. Moreover, since the norm of $\beta^*$ could be absorbed into $f$, we assume that $\|\beta^*\|_2 = 1$ for identifiability. Finally, we assume that $f$ and $X$ satisfy $\mathbb{E}[f''(\langle X, \beta^* \rangle)] > 0$.

Note that in Definition 2.3, we assume without any loss of generality that $\mathbb{E}[f''(\langle X, \beta^* \rangle)]$ is positive. If $\mathbb{E}[f''(\langle X, \beta^* \rangle)]$ is negative, one could replace $f$ by $-f$ by flipping the sign of $Y$. In another word, we essentially only require that $\mathbb{E}[f''(\langle X, \beta^* \rangle)]$ is nonzero. Intuitively, such a restriction on $f$ implies that the second-order cross-moments contains the information of $\beta^*$. Thus, we name this type of link functions as the second-order link. Similarly, we define MIM with second-order link.

**Definition 2.4** (MIM with Second-Order Link). The response $Y \in \mathbb{R}$ and the covariate $X \in \mathbb{R}^d$ are linked via

$$Y = f\left(\langle X, \beta_1^* \rangle, \ldots, \langle X, \beta_k^* \rangle\right) + \epsilon, \tag{2.4}$$

where $f \colon \mathbb{R}^k \to \mathbb{R}$ is an unknown link function, $\{\beta_\ell^*\}_{\ell \in [k]} \subseteq \mathbb{R}^d$ are the parameters of interest, and $\epsilon \in \mathbb{R}$ is the exogenous random noise that satisfies $\mathbb{E}(\epsilon) = 0$. In addition, we assume that the entries of $X$ are i.i.d. random variables with density $p_0$ and that $\{\beta_\ell^*\}_{\ell \in [k]}$ span a $k$-dimensional subspace of $\mathbb{R}^d$. Let $B^* = (\beta_1^* \ldots \beta_k^*) \in \mathbb{R}^{d \times k}$. The model in (2.4) could be reformulated as $Y = f(XB^*) + \epsilon$. By QR-factorization, we could write $B^*$ as $Q^* R^*$, where $Q^* \in \mathbb{R}^{d \times k}$ is an orthonormal matrix and $R^* \in \mathbb{R}^{k \times k}$ is invertible. Since $f$ is unknown, $R^*$ could be absorbed into the link function. Thus, we assume that $B^*$ is orthonormal for identifiability. We further assume that $B^*$ is $s^*$-row sparse, that is, $B^*$ contains only $s^*$ nonzero rows. Note that this definition of row sparsity does not depends on the choice of coordinate system. Finally, we assume that $f$ and $X$ satisfy $\lambda_{\min}(\mathbb{E}[\nabla^2 f(XB^*)]) > 0$.

In Definition 2.4, the assumption that $\mathbb{E}[\nabla^2 f(XB^*)]$ is positive definite is a multivariate generalization of the condition that $\mathbb{E}[f''(\langle X, \beta^* \rangle)] > 0$ for SIM in Definition 2.3. It essentially guarantees that estimating the projector of the subspace spanned by $\{\beta_\ell^*\}_{\ell \in [k]}$ is information-theoretically feasible.

# 3 Estimation Method and Main Results

We now introduce our estimators and establish their statistical rates of convergence. Discussion on the optimality of the established rates and connection to sparse PCA are deferred to §4. Recall that we focus on the case in which $X$ has i.i.d. entries with density $p_0 \colon \mathbb{R} \to \mathbb{R}$. Hence, the joint density of $X$ is $p(x) = p_0^{\otimes d}(x) = \prod_{j=1}^{d} p_0(x_j)$. For notational simplicity, let $s_0(u) = p_0'(u)/p_0(u)$. Then the first-order score function associated with $p$ is $S(x) = s_0 \circ (x)$. Equivalently, the $j$-th entry of the first-order score function associated with $p$ is given by $[S(x)]_j = s_0(x_j)$. Moreover, the second-order score function is

$$T(x) = S(x)S(x)^\top - \nabla S(x) = S(x)S(x)^\top - \mathrm{diag}\big[s_0' \circ (x)\big]. \tag{3.1}$$

Before we present our estimator, we introduce the assumption on $Y$ and $s_0(\cdot)$.

**Assumption 3.1** (Bounded Moment). We assume there exists a constant $M$ such that $\mathbb{E}_{p_0}[s_0(U)^6] \le M$ and $\mathbb{E}(Y^6) \le M$. We denote $\sigma_0^2 = \mathbb{E}_{p_0}[s_0(U)^2] = \mathrm{Var}_{p_0}[s_0(U)]$.

The assumption that $\mathbb{E}_{p_0}[s_0(U)^6] \le M$ allows for a broad family of distributions including Gaussian and more heavy-tailed random variables. Furthermore, we do not require the covariate to be elliptically symmetric as is commonly required in existing methods, which enables our estimator to be applicable for skewed covariates. As for the assumption that $\mathbb{E}(Y^6) \le M$, note that in the case of SIM we have

$$\mathbb{E}(Y^6) \le C\big(\mathbb{E}(\epsilon^6) + \mathbb{E}\big[f^6(\langle X, \beta^* \rangle)\big]\big).$$

Thus this assumption is satisfied as long as both $\epsilon$ and $f(\langle X, \beta^* \rangle)$ have bounded sixth moments. This is a mild assumption that allows for heavy-tailed response. Now we are ready to present our estimator for the sparse SIM in Definition 2.3. Recall that by Proposition 2.2 we have

$$\mathbb{E}\big[Y \cdot T(X)\big] = C_0 \cdot \beta^* \beta^{*\top}, \tag{3.2}$$

where $C_0 = 2\mathbb{E}[f''(\langle X, \beta^* \rangle)] > 0$ as in Definition 2.3. Hence, one way to estimator $\beta^*$ is to obtain the leading eigenvector of the sample version of $\mathbb{E}[Y \cdot T(X)]$. Moreover, as $\beta^*$ is sparse, we formulate our estimator as a semidefinite program

$$\begin{aligned} &\text{maximize } \big\langle W, \widetilde{\Sigma} \big\rangle - \lambda \|W\|_1 \\ &\text{subject to } 0 \preceq W \preceq I_d, \ \mathrm{Trace}(W) = 1. \end{aligned} \tag{3.3}$$

Here $\widetilde{\Sigma}$ is an estimator of $\Sigma^* = \mathbb{E}[Y \cdot T(X)]$, which is defined as follows. Note that both the score $T(X)$ and the response variable $Y$ could be heavy-tailed. In order to obtain near-optimal estimates in the finite-sample setting, we apply the truncation technique to handle the heavy-tails. In specific, for a positive threshold parameter $\tau \in \mathbb{R}$, we define the truncated random variables by

$$\widetilde{Y}_i = \mathrm{sign}(Y_i) \cdot \min\{|Y_i|, \tau\} \ \text{ and } \ \big[\widetilde{T}(X_i)\big]_{jk} = \mathrm{sign}\big\{T_{jk}(X_i)\big\} \cdot \min\big\{|T_{jk}(X_i)|, \tau^2\big\}. \tag{3.4}$$

Then we define the robust estimator of $\Sigma^*$ as

$$\widetilde{\Sigma} = \frac{1}{n} \sum_{i=1}^{n} \widetilde{Y}_i \cdot \widetilde{T}(X_i). \tag{3.5}$$

We denote by $\widehat{W}$ the solution of the convex optimization problem in (3.3), where $\lambda$ is a regularization parameter to be specified later. The final estimator $\widehat{\beta}$ is defined as the leading eigenvector of $\widehat{W}$. The following theorem quantifies the statistical rates of convergence of the proposed estimator.

**Theorem 3.2.** Let $\lambda = 10\sqrt{M \log d / n}$ in (3.3) and $\tau = (1.5Mn/\log d)^{1/6}$ in (3.4). Then under Assumption 3.1, we have $\|\widehat{\beta} - \beta^*\|_2 \le 4\sqrt{2}/C_0 \cdot s^* \lambda$ with probability at least $1 - d^{-2}$.

Now we introduce the estimator of $B^*$ for the sparse MIM in Definition 2.4. Proposition 2.2 implies that $\mathbb{E}[Y \cdot T(X)] = B^* D_0 B^*$, where $D_0 = \mathbb{E}[\nabla^2 f(X B^*)]$ is positive definite. Similar to (3.3), we recover the column space of $B^*$ by solving

$$\begin{aligned} &\text{maximize } \big\langle W, \widetilde{\Sigma} \big\rangle - \lambda \|W\|_1, \\ &\text{subject to } 0 \preceq W \preceq I_d, \ \mathrm{Trace}(W) = k, \end{aligned} \tag{3.6}$$

where $\widetilde{\Sigma}$ is defined in (3.5), $\lambda > 0$ is a regularization parameter, and $k$ is the number of indices, which is assumed to be known. Let $\widehat{W}$ be the solution of (3.6), and let the final estimator $\widehat{B}$ contain the top $k$ leading eigenvectors of $\widehat{W}$ as columns. For such an estimator, we have the following theorem quantifying its statistical rate of convergence. Let $\rho_0 = \lambda_{\min}(\mathbb{E}[\nabla^2 f(XB^*)])$.

**Theorem 3.3.** Let $\lambda = 10\sqrt{M \log d/n}$ in (3.6) and $\tau = (1.5Mn/\log d)^{1/6}$ in (3.4). Then under Assumption 3.1, with probability at least $1 - d^{-2}$, we have

$$\inf_{O \in \mathbb{O}_k} \left\| \widehat{B} - B^*O \right\|_{\mathrm{F}} \leq 4\sqrt{2}/\rho_0 \cdot s^* \lambda,$$

where $\mathbb{O}_k \in \mathbb{R}^{k \times k}$ is the set of all possible rotation matrices.

Minimax lower bounds for subspace estimation for MIM are established in [22]. For $k$ being fixed, Theorem 3.3 is near-optimal from a minimax point of view. The difference between the optimal rate and the above theorem is roughly a factor of $\sqrt{s^*}$. We will discuss more about this gap in Section 4. The proofs of Theorem 3.2 and Theorem 3.3 are provided in the supplementary material.

**Remark 3.4.** Recall that our discussion above is under the assumption that the entries of $X$ are i.i.d., which could be relaxed to the case of weak dependence between the covariates without any significant loss in the statistical rates presented above. We do not focus on this extension in this paper as we aim to clearly convey the main message of the paper in a simpler setting.

# 4 Optimality and Connection to Sparse PCA

Now we discuss the optimality of the results presented in §3. Throughout the discussion we assume that $k$ is fixed and does not increase with $d$ and $n$. The estimators for SIM in (3.3) and MIM in (3.6) are closely related to the semidefinite program-based estimator for sparse PCA [33]. In specific, let $X \in \mathbb{R}^d$ be a random vector with $\mathbb{E}(X) = 0$ and covariance $\Sigma = \mathbb{E}(XX^\top)$, which is symmetric and positive definite. The goal of sparse PCA is to estimate the projector onto the subspace spanned by the top $k$ eigenvectors, namely $\{v_\ell^*\}_{\ell \in [k]}$, of $\Sigma$, under the subspace sparsity assumption as specified in Definition 2.4. An estimator based on semidefinite programing is introduced in [33, 34], which is based on solving

$$\begin{aligned} &\text{maximize } \langle W, \widehat{\Sigma} \rangle - \lambda \|W\|_1 \\ &\text{subject to } 0 \preceq W \preceq I_d, \ \mathrm{Trace}(W) = k. \end{aligned} \tag{4.1}$$

Here $\widehat{\Sigma} = n^{-1} \sum_{i=1}^n X_i X_i^\top$ is the sample covariance matrix given $n$ i.i.d. observations $\{X_i\}_{i=1}^n$ of $X$. Note that the main difference between the SIM estimator and the sparse PCA estimator is the use of $\widetilde{\Sigma}$ in place of $\widehat{\Sigma}$. It is known that sparse PCA problem exhibits an interesting statistical-computational tradeoff [16, 34, 35], which naturally appears in the context of SIM as well. In particular, while the optimal statistical rate for sparse PCA is $\mathcal{O}(\sqrt{s^* \log d/n})$, the SDP-based estimator could only attain $\mathcal{O}(s^*\sqrt{\log d/n})$ under the assumption that $X$ is light-tailed. It is known that when $n = \Omega(s^{*2} \log d)$, one could obtain the optimal statistical rate of $\mathcal{O}(\sqrt{s^* \log d/n})$ by nonconvex method [37]. However, their results rely on the sharp concentration of $\widehat{\Sigma}$ to $\Sigma$ in the restricted operator norm:

$$\left\| \widehat{\Sigma} - \Sigma^* \right\|_{\mathrm{op},s} = \sup\left\{ w^\top (\widehat{\Sigma} - \Sigma) w : \|w\|_2 = 1, \|w\|_0 \leq s \right\} = \mathcal{O}(\sqrt{s \log d/n}). \tag{4.2}$$

When $X$ has heavy-tailed entries, for example, with bounded fourth moment, its highly unlikely that (4.2) holds.

**Heavy-tailed sparse PCA:** Recall that our estimators leverage a data-driven truncation argument to handle heavy-tailed distributions. Owing to the close relationship between our SIM/MIM estimators and the sparse PCA estimator, it is natural to ask whether such a truncation argument could lead to a sparse PCA estimator for heavy tailed $X$. Below we show it is indeed possible to obtain a near-optimal estimator for heavy-tailed sparse PCA based on the truncation technique. For vector $v \in \mathbb{R}^d$, let $\vartheta(v)$ be a truncation operator that works entrywise as $[\vartheta(v)]_j = \mathrm{sign}[v_j] \cdot \min\{|v_j|, \tau\}$ for $j \in [d]$. Then, our estimator is defined as follows,

$$\begin{aligned} &\text{maximize } \langle W, \overline{\Sigma} \rangle - \lambda \|W\|_1 \\ &\text{subject to } 0 \preceq W \preceq I_d, \ \mathrm{Trace}(W) = k, \end{aligned} \tag{4.3}$$

where $\overline{\Sigma} = n^{-1} \sum_{i=1}^{n} \overline{X}_i \overline{X}_i^\top$ and $\overline{X}_i = \vartheta(X_i)$, for $i = 1, \ldots n$. For the above estimator, we have the following theorem under the assumption that $X$ has heavy-tailed marginals. Let $V^* = (v_1^* \ldots v_k^*) \in \mathbb{R}^{d \times k}$ and we assume that $\rho_0 = \lambda_k(\Sigma) - \lambda_{k+1}(\Sigma) > 0$.

**Theorem 4.1.** Let $\widehat{W}$ be the solution of the optimization in (4.3) and let $\widehat{V} \in \mathbb{R}^{d \times k}$ contain the top $k$ leading eigenvectors of $\widehat{W}$. Also, we set the regularization parameter in (4.3) to be $\lambda = C_1 \sqrt{M \log d / n}$ and the truncation parameter to be $\tau = (C_2 M n / \log d)^{1/4}$, where $C_1$ and $C_2$ are some positive constants. Moreover, we assume that $V^*$ contains only $s^*$ nonzero rows and that $X$ satisfies $\mathbb{E}|X_j|^4 \leq M$ and $\mathbb{E}|X_i \cdot X_j|^2 \leq M$. Then, with probability at least $1 - d^{-2}$, we have

$$\inf_{O \in \mathbb{O}_k} \left\| \widehat{V} - V^* O \right\|_{\mathrm{F}} \leq 4\sqrt{2}/\rho_0 \cdot s^* \lambda,$$

where $\mathbb{O}_k \in \mathbb{R}^{k \times k}$ is the set of all possible rotation matrices.

The proof of the above theorem is identical to that of Theorem 3.3 and thus we omit it. The above theorem shows that with elementwise truncation, as long as $X$ satisfies a bounded fourth moment condition, the SDP estimator for sparse PCA achieves the near-optimal statistical rate of $\mathcal{O}(s^* \sqrt{\log d / n})$. We end this section with the following questions based on the above discussions:

1. Could we obtain the minimax optimal statistical rate $\mathcal{O}(\sqrt{s \log d / n})$ for sparse PCA in the high sample size regime with $n = \Omega(s^{*2} \log d)$ if $X$ has only bounded fourth moment?

2. Could we obtain the minimax optimal statistical rate $\mathcal{O}(\sqrt{s \log d / n})$ given $n = \Omega(s^{*2} \log d)$ when $f, X$, and $Y$ satisfy the bounded moment condition in Assumption 3.1 for MIM?

The answers to both questions lie in constructing truncation-based estimators that concentrate sharply in restricted operator norm defined in (4.2) or more realistically exhibit one-sided concentration bounds (see, e.g., [24] and [27] for related results and discussion). Obtaining such an estimator seems to be challenging for heavy-tailed sparse PCA and it is not immediately clear if this is even possible. We plan to report our findings for the above problem in the near future.

## 5 Experimental Results

In this section, we evaluate the finite-sample error of the proposed estimators on simulated data. We concentrate on the case of sparse phase retrieval. Recall that in this case, the link function is known and existing convex and non-convex estimators are applicable predominantly for the case of Gaussian or light-tailed data. The question of what are the necessary assumptions on the measurement vectors for (sparse) phase retrieval to work is an intriguing one [11]. In the sequel, we demonstrate that using the proposed score-based estimators, one could use heavy-tailed and skewed measurements as well, which significantly extend the class of measurement vectors applicable for sparse phase retrieval.

Recall that the covariate $X$ has i.i.d. entries with distribution $p_0$. Throughout this section, we set $p_0$ to be Gamma distribution with shape parameter 5 and scale parameter 1 or Rayleigh distribution with scale parameter 2. The random noise $\epsilon$ is set to be standard Gaussian. Moreover, we solve the optimization problems in (3.3) and (3.6) via the alternating direction method of multipliers (ADMM) algorithm, which introduces a dual variable to handle the constraints and updates the primal and dual variables iteratively.

For SIM, let the link functions be $f_1(u) = u^2$, $f_2 = |u|$, and $f_3(u) = 4u^2 + 3\cos(u)$, correspondingly. Here $f_1$ corresponds to the phase retrieval model and $f_2$ and $f_3$ could be viewed as its robust extension. Throughout the experiment we vary $n$ and fix $d = 500$ and $s^* = 5$. Also, the support of $\beta^*$ is chosen uniformly at random from all the possible subsets of $[d]$ with cardinality $s^*$. For each $j \in \mathrm{supp}(\beta^*)$, we set $\beta_j^* = 1/\sqrt{s^*} \cdot \gamma_j$, where $\gamma_j$'s are i.i.d. Rademacher random variables. Furthermore, we fix the regularization parameter $\lambda = 4\sqrt{\log d / n}$ and threshold parameter $\tau = 20$. In addition, we adopt the cosine distance $\cos \angle(\widehat{\beta}, \beta^*) = 1 - |\langle \widehat{\beta}, \beta^* \rangle|$, to measure the estimation error. We plot the cosine distance against the theoretical statistical rate of convergence $s^* \sqrt{\log d / n}$ in Figure 2-(a)-(c) for each link function, respectively. The plot is based on 100 independent trials for each $n$. It shows that the estimation error is bounded by a linear function of $s^* \sqrt{\log d / n}$, which corroborates the theory.

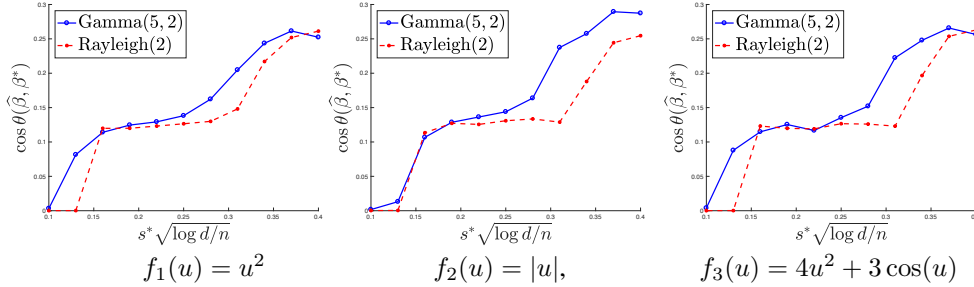

$$f_1(u) = u^2 \qquad f_2(u) = |u|, \qquad f_3(u) = 4u^2 + 3\cos(u)$$

Figure 2: Cosine distances between the true parameter $\beta^*$ and the estimated parameter $\widehat{\beta}$ in the sparse SIM with the link function in one of $f_1$, $f_2$, and $f_3$. Here we set $d = 500$. $s^* = 5$ and vary $n$.

## 6 Discussion

In this work, we study estimating the parametric component of SIM and MIM in the high dimensions, under fairly general assumptions on the link function $f$ and response $Y$. Furthermore, our estimators are applicable in the non-Gaussian setting in which $X$ is not required to satisfy restrictive Gaussian or elliptical symmetry assumptions. Our estimators are based on a data-driven truncation technique in combination with a second-order Stein's identity.

In the low-dimensional setting, for two-layer neural network [14] propose a tensor-based method for estimating the parametric component. Their estimators are sub-optimal even assuming $X$ is Gaussian. An immediate application of our truncation-based estimators enables us to obtain optimal results for a fairly general class of covariate distributions in the low-dimensional setting. Obtaining optimal or near-optimal results in the high-dimensional setting is of great interest for two-layer neural network, albeit challenging. We plan to extend the results of the current paper to two-layer neural networks in high dimensions and report our findings in the near future.

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
