[Supplementary Material 1]

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

# A  Proof of the Main Results

In this section we prove the main results. We first prove that the proposed estimators achieve near-optimal statistical rates of convergence. Then we prove the supporting lemma on our data-driven approach of truncation.

## A.1  Proof of Theorem 3.2

*Proof.* We denote by $\widehat{W}$ the solution of the convex program in (3.3). Also, let $W^* = \beta^* \beta^{*\top}$. In the following, we establish an upper bound for $\|\widehat{W} - W^*\|_2$.

Since $W^*$ is feasible for the optimization problem in (3.3), we have

$$\langle \widehat{W}, \widetilde{\Sigma} \rangle - \lambda \|\widehat{W}\|_1 \geq \langle W^*, \widetilde{\Sigma} \rangle - \lambda \|W^*\|_1. \tag{A.1}$$

We denote $\Sigma^* = \mathbb{E}[Y \cdot T(X)]$. Note that $\beta^*$ is the leading eigenvector of $\Sigma^*$. Then (A.1) is equivalent to

$$\langle \widehat{W} - W^*, \widetilde{\Sigma} - \Sigma^* \rangle - \lambda \|\widehat{W}\|_1 + \lambda \|W^*\|_1 \geq \langle \Sigma^*, W^* - \widehat{W} \rangle. \tag{A.2}$$

The following Lemma in [33] establishes an upper bound for the first term on the left-hand side of (A.2).

**Lemma A.1.** Let $\Omega \in \mathbb{R}^{d \times d}$ be a symmetric matrix and let $\lambda_1 \geq \lambda_2 \geq \ldots \lambda_d$ the eigenvalues of $\Omega$ in descending order. For any $\ell \in [d-1]$ such that $\lambda_\ell - \lambda_{\ell+1} > 0$, let $\Pi_\ell \in \mathbb{R}^{d \times d}$ be the projection matrix for the subspace spanned by the eigenvectors of $\Omega$ corresponding to $\lambda_1, \ldots, \lambda_\ell$. Then for any $\Lambda \in \mathbb{R}^{d \times d}$ satisfying $0 \preceq \lambda \preceq I_d$ and $\mathrm{Trace}(\Lambda) = k$, we have

$$(\lambda_\ell - \lambda_{\ell+1}) \cdot \|\Pi_k - \Lambda\|_\mathrm{F}^2 \leq 2\langle \Omega, \Pi_k - \Lambda \rangle.$$

Note that $W^*$ is the projection matrix for the subspace spanned by $\beta^*$. Applying Lemma A.1 to $\Sigma^*$ with $\ell = 1$, we have

$$\langle \Sigma^*, W^* - \widehat{W} \rangle \geq C_0/2 \cdot \|\widehat{W} - W^*\|_\mathrm{F}^2, \tag{A.3}$$

where $C_0 > 0$ is defined in (3.2). In addition, by Hölder's inequality, we have

$$\langle \widehat{W} - W^*, \widetilde{\Sigma} - \Sigma^* \rangle \leq \|\widetilde{\Sigma} - \Sigma^*\|_\infty \cdot \|\widehat{W} - W^*\|_1. \tag{A.4}$$

In what follows, we bound $\|\widetilde{\Sigma} - \Sigma^*\|_\infty$.

**Lemma A.2.** Let $\widetilde{\Sigma}$ be defined in (3.5) and we define $\Sigma^* = \mathbb{E}[Y \cdot T(X)]$. Under Assumption 3.1, for any truncation level $\tau > 0$ in (3.4), with probablity at least $1 - d^{-2}$, we have

$$\|\widetilde{\Sigma} - \Sigma^*\|_\infty \leq 9M \cdot \tau^{-3} + 2\tau^3 \cdot \log d/n + 2\sqrt{5M \cdot \log d/n}. \tag{A.5}$$

*Proof.* See §A.3 for a detailed proof. $\qquad \square$

By this lemma, if we set $\tau = (1.5Mn/\log d)^{1/6}$, then with probability at least $1 - d^{-1}$,

$$\|\widetilde{\Sigma} - \Sigma^*\|_\infty \leq (2\sqrt{5} + 2\sqrt{6}) \cdot \sqrt{M \log d/n} \leq 10\sqrt{M \log d/n}. \tag{A.6}$$

Thus by setting $\lambda = 10\sqrt{M \log d/n}$ we have $\|\widetilde{\Sigma} - \Sigma^*\|_\infty \leq \lambda$ with probability at least $1 - d^{-2}$.

Then combining (A.2), (A.3), and (A.4) we have

$$\lambda \left( \|\widehat{W} - W^*\|_1 - \|\widehat{W}\|_1 + \|W^*\|_1 \right) \geq C_0/2 \cdot \|\widehat{W} - W^*\|_\mathrm{F}^2. \tag{A.7}$$

Note that $W^* = \beta^* \beta^{*\top}$ and that $\beta^*$ is $s^*$-sparse. We denote the support of $W^*$ by $\mathcal{J}$, which is given by

$$\mathcal{J} = \left\{ (j,k) \in [d] \times [d] \colon \beta_j^* \cdot \beta_k^* \neq 0 \right\}.$$

Then by seperation of the $\ell_1$-norm, we have

$$\|\widehat{W}\|_1 = \|\widehat{W}_\mathcal{J}\|_1 + \|\widehat{W}_{\mathcal{J}^c}\|_1 \quad \text{and} \quad \|\widehat{W} - W^*\|_1 = \|\widehat{W}_\mathcal{J} - W_\mathcal{J}^*\|_1 + \|\widehat{W}_{\mathcal{J}^c}\|_1,$$

which implies that

$$\|\widehat{W} - W^*\|_1 - \|\widehat{W}\|_1 + \|W^*\|_1 = \|\widehat{W}_{\mathcal{J}} - W^*_{\mathcal{J}}\|_1 + \|\widehat{W}_{\mathcal{J}}\|_1 - \|W^*_{\mathcal{J}}\|_1$$
$$\leq 2\|\widehat{W}_{\mathcal{J}} - W^*_{\mathcal{J}}\|_1 \leq 2s^*\|\widehat{W} - W^*\|_{\mathrm{F}}. \tag{A.8}$$

Here the last inequality in (A.8) follows from the fact that $|\mathcal{J}| = s^{*2}$. Combining (A.7) and (A.8), we obtain

$$\|\widehat{W} - W^*\|_{\mathrm{F}} \leq 4/C_0 \cdot s^*\lambda. \tag{A.9}$$

Since $\widehat{\beta}$ is the leading eigenvector of $\widehat{W}$, we have $\|\widehat{\beta} - \beta^*\|_2 \leq \sqrt{2}\|\widehat{W} - W^*\|_{\mathrm{F}} \leq 4\sqrt{2}/C_0 \cdot s^*\lambda$, which concludes the proof. $\qquad\square$

## A.2 Proof of Theorem 3.3

*Proof.* The proof is similar to that of Theorem 3.2. In the case of sparse MIM, we denote $W^* = B^*B^{*\top}$. Note that $\widehat{W}$ is the solution to the optimization problem in (3.6) and that $\widehat{B}$ consists of the top $k$ leading eigenvectors of $\widehat{W}$. Then by Corollary 3.2 in [33], we have

$$\inf_{O \in \mathbb{O}_k} \|\widehat{B} - B^*O\|_{\mathrm{F}} \leq \sqrt{2}\|\widehat{W} - W^*\|_{\mathrm{F}}. \tag{A.10}$$

In what follows, we derive an upper bound for $\widehat{W} - W^*$. Since $B^*$ is orthonormal, $\mathrm{Trace}(W^*) = k$. Thus $W^*$ is feasible for (3.6), which implies

$$\langle \widehat{W} - W^*, \widetilde{\Sigma} - \Sigma^* \rangle - \lambda\|\widehat{W}\|_1 + \lambda\|W^*\|_1 \geq \langle \Sigma^*, W^* - \widehat{W} \rangle. \tag{A.11}$$

Here we define $\Sigma^* = \mathbb{E}[Y \cdot T(X)]$. Note that $W^*$ is the projection matrix for the subspace spanned by the top-$k$ leading eigenvectors of $\Sigma^*$. By Lemma A.1 with $\ell = k$, we have

$$\langle \Sigma^*, W^* - \widehat{W} \rangle \geq \rho_0/2 \cdot \|\widehat{W} - W^*\|_{\mathrm{F}}^2,$$

where $\rho_0$ is the smallest eigenvalue of $\mathbb{E}[\nabla^2 f(XB^*)]$. Similar to the proof of Theorem 3.2, by using Hölder's inequality and (A.11), we have

$$\|\widetilde{\Sigma} - \Sigma^*\|_\infty \cdot \|\widehat{W} - W^*\|_1 - \lambda\|\widehat{W}\|_1 + \lambda\|W^*\|_1 \geq \rho_0/2 \cdot \|\widehat{W} - W^*\|_{\mathrm{F}}^2. \tag{A.12}$$

By Lemma A.2, if we set $\lambda = 10\sqrt{M\log d/n}$, with probability at least $1 - d^{-2}$, we have

$$\|\widehat{\Sigma} - \Sigma^*\|_\infty \leq \lambda. \tag{A.13}$$

Note that the support of $W^*$ is

$$\mathcal{J} \subseteq \big\{(j,k) \in [d] \times [d] \colon \|B^*_{j\cdot}\|_2 \cdot \|B^*_{k\cdot}\|_2 \neq 0\big\}.$$

Since $B^*$ is $s^*$-row sparse, $|\mathcal{J}| \leq s^{*2}$. Thus (A.8) also hold for the MIM. Combining (A.12), (A.13), and (A.8), we obtain

$$\|\widehat{W} - W^*\|_{\mathrm{F}} \leq 4/\rho_0 \cdot s^*\lambda. \tag{A.14}$$

Finally, combining (A.10) and (A.14), we conclude the proof. $\qquad\square$

## A.3 Proof of Lemma A.2

*Proof.* By triangle inequailty, we have

$$\|\widetilde{\Sigma} - \Sigma^*\|_\infty \leq \|\widetilde{\Sigma} - \mathbb{E}\widetilde{\Sigma}\|_\infty + \|\mathbb{E}\widetilde{\Sigma} - \Sigma^*\|_\infty. \tag{A.15}$$

In the sequel, we bound the second term on the right-hand side of (A.15), which controls the bias of truncation. For each $j, k \in [d]$, we have

$$\left|\mathbb{E}\widetilde{\Sigma}_{jk} - \Sigma^*_{jk}\right| \leq \left|\mathbb{E}\big[\widetilde{Y} \cdot \widetilde{T}_{jk}(X)\big] - \mathbb{E}\big[Y \cdot T_{jk}(X)\big]\right|$$
$$\leq \left|\mathbb{E}\big\{\widetilde{Y} \cdot \big[\widetilde{T}_{jk}(X) - T_{jk}(X)\big]\big\}\right| + \left|\mathbb{E}\big[(\widetilde{Y} - Y) \cdot T_{jk}(X)\big]\right|. \tag{A.16}$$

For the first term in (A.16), note that $\widetilde{T}_{jk}(X) - T_{jk}(X) = T_{jk}(X) \cdot \mathbb{1}\{|T_{jk}(X)| \geq \tau^2\}$. Then by Cauchy-Schwarz inequality we have

$$\left| \mathbb{E}\left\{ \widetilde{Y} \cdot [\widetilde{T}_{jk}(X) - T_{jk}(X)] \right\} \right|^2 = \left| \mathbb{E}\left[ \widetilde{Y} \cdot T_{jk}(X) \cdot \mathbb{1}\{|T_{jk}(X)| \geq \tau^2\} \right] \right|^2$$
$$\leq \mathbb{E}[\widetilde{Y}^2 \cdot T_{jk}^2(X)] \cdot \mathbb{P}\left[ |T_{jk}(X)| \geq \tau^2 \right]. \tag{A.17}$$

Furthermore, by Hölder's inequality, we have

$$\mathbb{E}[\widetilde{Y}^2 \cdot T_{jk}^2(X)] \leq [\mathbb{E}(\widetilde{Y}^6)]^{1/3} \cdot \left\{ \mathbb{E}[|T_{jk}(X)|^3] \right\}^{2/3} \leq [\mathbb{E}(Y^6)]^{1/3} \left\{ \mathbb{E}[|T_{jk}^3(X)|] \right\}^{2/3}. \tag{A.18}$$

If $j \neq k$, by the definition of $T(x)$ in (3.1), we have $T_{jk}(x) = S_j(x) \cdot S_k(x), \forall x \in \mathbb{R}^d$. Then by Cauchy-Schwarz inequality, we have

$$\mathbb{E}\left[ |T_{jk}^3(X)| \right] = \mathbb{E}\left[ |S_j(X)|^3 \cdot |S_k(X)|^3 \right] \leq \sqrt{\mathbb{E}[S_j^6(X)] \cdot \mathbb{E}[S_k^6(X)]} = \mathbb{E}[S_j^6(X)]. \tag{A.19}$$

In addition, if $j = k$, by (3.1), $T_{jj}(x) = S_j^2(x) - s_1(x_j)$. Since $(a+b)^3 \leq 4(a^3 + b^3)$ for any $a, b > 0$, we have

$$\mathbb{E}\left[ |T_{jj}^3(X)| \right] \leq 4\mathbb{E}[S_j^6(X)] + 4\mathbb{E}\left[ |s_1^3(X_j)| \right]. \tag{A.20}$$

Moreover, by (A.17), (A.18), and the Markov's inequality

$$\mathbb{P}\left[ |T_{jk}(X)| \geq \tau^2 \right] \leq \mathbb{E}\left[ |T_{jk}^3(X)| \right] \cdot \tau^{-6},$$

we further have

$$\left| \mathbb{E}\left\{ \widetilde{Y} \cdot [\widetilde{T}_{jk}(X) - T_{jk}(X)] \right\} \right|^2 \leq [\mathbb{E}(Y^6)]^{1/3} \cdot \left\{ \mathbb{E}\left[ |T_{jk}^3(X)| \right] \right\}^{5/3} \cdot \tau^{-6} \leq 32M^2 \cdot \tau^{-6}. \tag{A.21}$$

Here the last inequality follows from combining Assumption 3.1, (A.19), and (A.20).

Similarly, for the second term in (A.16), by the Hölder's inequality and the Markov's inequality we obtain

$$\left| \mathbb{E}[(\widetilde{Y} - Y) \cdot T_{jk}(X)] \right|^2 \leq [\mathbb{E}(Y^6)]^{1/3} \cdot \left\{ \mathbb{E}\left[ |T_{jk}^3(X)| \right] \right\}^{2/3} \cdot \mathbb{P}(|Y| \geq \tau)$$
$$\leq [\mathbb{E}(Y^6)]^{4/3} \cdot \left\{ \mathbb{E}\left[ |T_{jk}^3(X)| \right] \right\}^{2/3} \cdot \tau^{-6} \leq 4M^2 \cdot \tau^{-6}. \tag{A.22}$$

Therefore, combining (A.16), (A.21), and (A.22), we obtain

$$\|\mathbb{E}\widetilde{\Sigma} - \Sigma^*\|_\infty \leq 9M \cdot \tau^{-3}. \tag{A.23}$$

In what follows, we give a high-probability bound on $\|\widetilde{\Sigma} - \mathbb{E}\widetilde{\Sigma}\|_\infty$ using concentration inequalities, which combined with A.23, concludes the proof.

For any $j, k \in [d]$, note that $|\widetilde{Y} \cdot \widetilde{T}_{jk}(X)| \leq \tau^3$. In addition, by assumption 3.1, its variance is bounded by

$$\mathrm{Var}[\widetilde{Y} \cdot \widetilde{T}_{jk}(X)] \leq \mathbb{E}\left[ Y^2 \cdot T_{jk}^2(X) \right] \leq [\mathbb{E}(Y^6)]^{1/3} \cdot \left\{ \mathbb{E}\left[ |T_{jk}^3(X)| \right] \right\}^{2/3} \leq 2M.$$

Now we apply the Bernstein's inequality [2] (Theorem 2.10) to $\{\widetilde{Y}_i \cdot \widetilde{T}_{jk}(X_i)\}_{i \in [n]}$ and obtain that

$$\mathbb{P}\left\{ \left| \frac{1}{n} \sum_{i=1}^n \widetilde{Y}_i \cdot \widetilde{T}_{jk}(X_i) - \mathbb{E}[\widetilde{Y} \cdot \widetilde{T}_{jk}(X)] \right| \geq \sqrt{\frac{4M \cdot t}{n}} + \frac{\tau^3 \cdot t}{3n} \right\} \leq 2\exp(-t). \tag{A.24}$$

Taking a union bound over $j, k \in [d]$ in (A.24), we obtain that

$$\mathbb{P}\left[ \|\widetilde{\Sigma} - \mathbb{E}\widetilde{\Sigma}\|_\infty \geq \sqrt{4M \cdot t/n} + \tau^3 \cdot t/(3n) \right] \leq 2\exp(-t + 2\log d). \tag{A.25}$$

Choosing $t = 5\log d$ in (A.25), we have

$$\|\widetilde{\Sigma} - \mathbb{E}\widetilde{\Sigma}\|_\infty \leq 2\sqrt{5M \log d/n} + 2\tau^3 \cdot \log d/n \tag{A.26}$$

with probablity at least $1 - d^{-2}$. Finally, combining (A.23) and (A.26), we conclude the proof of Lemma A.2. $\qquad \square$

[Supplementary Material 2]

# A  Proof of the Main Results

In this section we prove the main results. We first prove that the proposed estimators achieve near-optimal statistical rates of convergence. Then we prove the supporting lemma on our data-driven approach of truncation.

## A.1  Proof of Theorem 3.2

*Proof.* We denote by $\widehat{W}$ the solution of the convex program in (3.3). Also, let $W^* = \beta^* \beta^{*\top}$. In the following, we establish an upper bound for $\|\widehat{W} - W^*\|_2$.

Since $W^*$ is feasible for the optimization problem in (3.3), we have

$$\langle \widehat{W}, \widetilde{\Sigma} \rangle - \lambda \|\widehat{W}\|_1 \geq \langle W^*, \widetilde{\Sigma} \rangle - \lambda \|W^*\|_1. \tag{A.1}$$

We denote $\Sigma^* = \mathbb{E}[Y \cdot T(X)]$. Note that $\beta^*$ is the leading eigenvector of $\Sigma^*$. Then (A.1) is equivalent to

$$\langle \widehat{W} - W^*, \widetilde{\Sigma} - \Sigma^* \rangle - \lambda \|\widehat{W}\|_1 + \lambda \|W^*\|_1 \geq \langle \Sigma^*, W^* - \widehat{W} \rangle. \tag{A.2}$$

The following Lemma in [33] establishes an upper bound for the first term on the left-hand side of (A.2).

**Lemma A.1.** Let $\Omega \in \mathbb{R}^{d \times d}$ be a symmetric matrix and let $\lambda_1 \geq \lambda_2 \geq \ldots \lambda_d$ the eigenvalues of $\Omega$ in descending order. For any $\ell \in [d-1]$ such that $\lambda_\ell - \lambda_{\ell+1} > 0$, let $\Pi_\ell \in \mathbb{R}^{d \times d}$ be the projection matrix for the subspace spanned by the eigenvectors of $\Omega$ corresponding to $\lambda_1, \ldots, \lambda_\ell$. Then for any $\Lambda \in \mathbb{R}^{d \times d}$ satisfying $0 \preceq \lambda \preceq I_d$ and $\mathrm{Trace}(\Lambda) = k$, we have

$$(\lambda_\ell - \lambda_{\ell+1}) \cdot \|\Pi_k - \Lambda\|_{\mathrm{F}}^2 \leq 2 \langle \Omega, \Pi_k - \Lambda \rangle.$$

Note that $W^*$ is the projection matrix for the subspace spanned by $\beta^*$. Applying Lemma A.1 to $\Sigma^*$ with $\ell = 1$, we have

$$\langle \Sigma^*, W^* - \widehat{W} \rangle \geq C_0/2 \cdot \|\widehat{W} - W^*\|_{\mathrm{F}}^2, \tag{A.3}$$

where $C_0 > 0$ is defined in (3.2). In addition, by Hölder's inequality, we have

$$\langle \widehat{W} - W^*, \widetilde{\Sigma} - \Sigma^* \rangle \leq \|\widetilde{\Sigma} - \Sigma^*\|_\infty \cdot \|\widehat{W} - W^*\|_1. \tag{A.4}$$

In what follows, we bound $\|\widetilde{\Sigma} - \Sigma^*\|_\infty$.

**Lemma A.2.** Let $\widetilde{\Sigma}$ be defined in (3.5) and we define $\Sigma^* = \mathbb{E}[Y \cdot T(X)]$. Under Assumption 3.1, for any truncation level $\tau > 0$ in (3.4), with probablity at least $1 - d^{-2}$, we have

$$\|\widetilde{\Sigma} - \Sigma^*\|_\infty \leq 9M \cdot \tau^{-3} + 2\tau^3 \cdot \log d/n + 2\sqrt{5M \cdot \log d/n}. \tag{A.5}$$

*Proof.* See §A.3 for a detailed proof. $\qquad\square$

By this lemma, if we set $\tau = (1.5 M n / \log d)^{1/6}$, then with probability at least $1 - d^{-1}$,

$$\|\widetilde{\Sigma} - \Sigma^*\|_\infty \leq (2\sqrt{5} + 2\sqrt{6}) \cdot \sqrt{M \log d/n} \leq 10\sqrt{M \log d/n}. \tag{A.6}$$

Thus by setting $\lambda = 10\sqrt{M \log d/n}$ we have $\|\widetilde{\Sigma} - \Sigma^*\|_\infty \leq \lambda$ with probability at least $1 - d^{-2}$.

Then combining (A.2), (A.3), and (A.4) we have

$$\lambda \left( \|\widehat{W} - W^*\|_1 - \|\widehat{W}\|_1 + \|W^*\|_1 \right) \geq C_0/2 \cdot \|\widehat{W} - W^*\|_{\mathrm{F}}^2. \tag{A.7}$$

Note that $W^* = \beta^* \beta^{*\top}$ and that $\beta^*$ is $s^*$-sparse. We denote the support of $W^*$ by $\mathcal{J}$, which is given by

$$\mathcal{J} = \left\{ (j, k) \in [d] \times [d] : \beta_j^* \cdot \beta_k^* \neq 0 \right\}.$$

Then by seperation of the $\ell_1$-norm, we have

$$\|\widehat{W}\|_1 = \|\widehat{W}_{\mathcal{J}}\|_1 + \|\widehat{W}_{\mathcal{J}^c}\|_1 \quad \text{and} \quad \|\widehat{W} - W^*\|_1 = \|\widehat{W}_{\mathcal{J}} - W_{\mathcal{J}}^*\|_1 + \|\widehat{W}_{\mathcal{J}^c}\|_1,$$