[Reviews · NeurIPS 2017]

Reviewer 1



Summary: This article studies the estimation problem of single index and multiple index models in high dimensional setting. Using the formulation of second order Stein's lemma with sparsity assumptions, they propose an estimator formulated as a solution to a sparsity constrained semi-definite programming. The statistical rate of convergence of the estimator is derived and a numerical example is given. Overall, I find the article interesting and useful, but their emphasis on "heavy tail" case and also partly non-Gaussianity is rather intriguing. Looking at the details, the main contribution the authors claim seems to be related to an extension of an optimality result for the estimation of sparse PCA from sub-Gaussian case to essentially sub-Exponential case through a Bernstein inequality. It would be helpful to clarify this point better. Detailed comments are summarized below. (a) Stein's lemma: I think one of the first references to an estimator based on Stein's lemma is H\"ardle and Stoker (1989), using the so-called "method of average derivatives", which does not require Gaussianity. (b) Derivative condition: Normally the condition on the second derivative would be stronger than that on the first derivative, yet, the article suggests the other way around. Is there any other place to pay the price, or does it really provide more general framework? (c) Heavy tail case: The notation of "heavy-tail" (as a pareto-like tail to most readers, I suspect, e.g. Barthe et al., 2005) is confusing and misleading in this context. Especially, I would think the heavy tail problem in economics has more to do with extremes and I doubt that the new approach is applicable in that context. Hence, I would suggest to remove the reference to the heavy-tail part, and instead make a better link to the contributions in terms of a relaxation of the tail conditions. (d) Moment condition: In fact, I wonder how the moment condition on the score function could be translated in terms of the standard moment conditions (and tail conditions) on the original random variables. It would be helpful to demonstrate this point, and perhaps give a condition to make those assumptions comparable. (e) Relation to Sparse PCA: Given the proximity to the formulation of the estimator to sparse PCA, I was imagining that a similar technique of the proofs from sparse PCA would have been used here, (except utilizing another (Berstein-type) concentration inequality), however, the relation to sparse PCA was presented as a consequence of their results. Then, in addition to the similarity to sparse PCA, could you also demonstrate a fundamental difference in techniques/considerations, if any, (other than the moment condition, or to overcome the moment condition?) required to deal with estimation problems of single index and multiple index models? (f) Numerical algorithm: The algorithm to compute the estimator is mentioned only in Section 5. I would suggest to include some computational details in Section 3 as well, perhaps after (3.6). (g) Minimum eigenvalue condition: I suppose without sparsity assumption, the minimum eigenvalue (line 174, p.5) should be zero for high dimensional case when d > k. Then, instead of "Finally" to introduce this condition, which seems to suggest that they are unrelated, would the connection need to be emphasized here as well? Minor comments and typos: (i) line 54, p2: (... pioneered by Ker-Chau Li, ...): From the flow of the arguments, it would be natural to add the references [20,21,22] here. (ii) line 75, p2: I understood "SDP" as "semi-definite programming" in the end, but it would be better to be clear from the beginning here. (iii) equation (4.1) p.4: typo. (iv) heck references: the style of references is not coherent. References: Barthe, F., Cattiaux, P. and Roberto, C. (2005) Concentration for independent random variables with heavy tails, Applied Mathematics Research Express (2). 39-60. H\"ardle, W. and Stoker, T.M. (1989) Investigating smooth multiple regression by the method of average derivatives, JASA 84 (408), 986-995.

Reviewer 2



This paper considers the estimation of semi-parametric sparse models, with possibly non-Gaussian (but random) covariates. This can be applied to non-linear sparse inverse problems with non-Gaussian sensing operators, which seems of particular interest in several compressed sensing problems. The new estimator is defined as the solution of a convex problem, and statistical convergence rates are proven, based on the second-order Stein identity. I must say that the paper is far from my research area, so I could not get too much into details. But it tackles an important issue, and it seems to me that the paper proposes a novel and valuable contribution, while being well written and well presented.

Reviewer 3



This paper studies estimation of sparse single and multiple index models. The goal is to construct good estimators under weaker assumptions on the covariate distribution and link function. This paper is poorly written. In the introduction the author(s) emphasized on the generality of their methods, such as how it would work for a wide range of covariate distributions and link functions. But in the main methodology section (Section 3), the development focused only on covariate distributions with independent, identically distributed components, which is far more restrictive than almost any existing works. Equation 3.2 only holds when the link function is phase retrieval. So the entire section 3 is restricted to a highly restrictive covariate distribution and a single link function. It is entirely unclear how the method works in other scenarios. In the introduction it is mentioned repeatedly that the proposed method needs to assume that the covariate distribution is known. It is unclear how this assumption is used in the methodological and theoretical developments. But this is a very strong assumption. For example, in high dimensional linear regression, if the covariate distribution is known, then the problem becomes almost trivial. The author(s) also claimed to make a contribution to heavy-tailed sparse PCA, which is over-stating. The only contribution in Section 4 is the use of a robust estimator of the input matrix $\bar\Sigma$. The robust estimator itself is not new. The SPCA formulation in eqs. (3.7) and (4.1) is standard and well-known (d'Aspremont et al SIAM Review 2007, Vu et al NIPS 2013), which the authors were aware of but did not cite in Sections 3 and 4. Given the existing results on sparse PCA analysis and truncated average estimate, the consistency of heavy-tailed sparse PCA is trivial: the input matrix is entry-wise consistent and hence $\ell_1$ penalized sparse PCA is consistent. Typos: - line 44: "statistical optimal" --> "statistically optimal" - line 52: "is rather restrictive" --> "are rather restrictive" - Equation 2.1: Remove the second "=". - First line of Eq. 3.3 (also eqs 3.7 and 4.1): "+" --> "-", unless you use a negative value of $\lambda$. ####### Response after author feedback ######## Thanks for the feedback. Yes. I agree that there was some misunderstanding in my first review. But still section 4.1 is simply a special case of the analysis of sparse PCA which only assumes that the input matrix is entry-wise consistent. This point has been illustrated, for example, in the perspective of robust optimization in d'Aspremont et al and later developments (for example, the proof used in Wang et al arXiv:1307.0164).